# Vitamin D: Before, during and after Pregnancy: Effect on Neonates and Children

**DOI:** 10.3390/nu14091900

**Published:** 2022-05-01

**Authors:** José Luis Mansur, Beatriz Oliveri, Evangelina Giacoia, David Fusaro, Pablo René Costanzo

**Affiliations:** 1Endocrinology and Metabolism Center, La Plata B1902ADQ, Argentina; 2Osteoporosis and Metabolic Bone Diseases Laboratory, Institute of Immunology, Genetics, and Metabolism (INIGEM), School of Pharmacy and Biochemistry, Hospital de Clinicas, University of Buenos Aires, Mautalen, Health and Research, Buenos Aires C1053ABH, Argentina; beatrizoliveri258@gmail.com; 3Endocrinology Service Hospital Posadas, University of Buenos Aires, El Palomar C1053ABH, Argentina; dragiacoia@gmail.com; 4Buenos Aires Gynecological Institute ACOG, Service Fundación Favaloro, Buenos Aires C1053ABH, Argentina; djfusaro@yahoo.com.ar; 5Andrology Section, Endocrinology, Metabolism and Nuclear Medicine Service, Hospital Italiano de Buenos Aires, Buenos Aires PC C1199ABB, Argentina

**Keywords:** Vitamin D, pregnancy, fertility

## Abstract

A worldwide high prevalence of vitamin D (VD) deficiency has become of growing concern because of potential adverse effects on human health, including pregnant women and their offsprings. Beyond its classical function as a regulator of calcium and phosphate metabolism, together with its fundamental role in bone health in every stage of life, its deficiency has been associated to multiple adverse health effects. The classic effects of VD deficiency in pregnancy and neonates have been late hypocalcemia and nutritional rickets. Nevertheless, recent studies have linked VD to fertility and 25(OH)D with several clinical conditions in pregnancy: preeclampsia, gestational diabetes, higher incidence of cesarean section and preterm birth, while in infants, the clinical conditions are low birth weight, lower bone mass and possible relationship with the development of such diseases as bronchiolitis, asthma, type 1 diabetes, multiple sclerosis and autism included as VD non-classical actions. The supplementation with Vitamin D and achievement of optimal levels reduce maternal-fetal and newborn complications. Supplementation in children with VD deficiency reduces the risk of respiratory infections and possibly autoimmune diseases and autism. This review emphasizes the roles of Vitamin D deficiency and the consequences of intervention from preconception to infancy.

## 1. Introduction

In most of the world populations, a high prevalence of Vitamin D deficiency is acknowledged in every stage of life, representing a current Public Health concern. The only sources of Vitamin D are exposure to the sun (UV radiation) and food, but only very few foods contain significant quantity. Vitamin D circulates bound to the Vitamin D binding protein (DBP), which is hydroxylated in the liver at 25-hydroxyvitamin D (25(OH)D), and then in the kidney by 1 alpha hydroxylase at 1-25 dihydroxyvitamin D (1-25 (OH)2D3), which is the active metabolite. There is also the production of active Vitamin D in other organs other than the kidney. The Vitamin D receptor (VDR), the enzyme 1 alpha hydroxylase and the production of 1-25 (OH)2D3 have been demonstrated in a variety of tissues. In addition to the classic impact in phospho-calcium homeostasis and in bone health, Vitamin D enables a very selective regulation of genes involved in the cardiovascular system processes, glucose metabolism, cell differentiation and immunoregulation. These observational studies show an inverse relationship between 25(OH)D and the prevalence of diseases but do not provide evidence of causality [1,2]. Meta-analyses with Vitamin D supplementation have shown conflictive results, particularly due to variables such as baseline 25(OH)D, sample size, different doses and Vitamin D schemes used, adherence to dosing regime and lack of measurement of 25(OH)D levels through trials, among other issues [3]. 

There is a controversy concerning the optimum levels of 25(OH)D [4,5]. 25(OH)D levels <20 ng/mL are considered an indication of Vitamin D deficiency, though the Endocrine Society and other groups of experts consider insufficiency with levels between 25(OH)D 20–29 ng/mL and ≥30 ng/mL as sufficient levels; for non-classical actions, some authors propose reaching >40 ng/mL levels.

In recent years, the evidence concerning the relevance of Vitamin D has increased, not only during the gestation stage but also before, due to its potential role in fertility. Furthermore, several studies have shown a high prevalence of Vitamin D deficiency in pregnant women all over the world, in every pregnancy trimester [6,7,8,9].

Within the determining factors of the concentration of 25(OH)D in pregnant women, the following has been described: skin pigmentation, UV radiation, extensive skin covering for religious or cultural reasons and greater social deprivation. The combination of pregnancy during winter months and obesity (body mass index >30 kg/m^2^) in the pregnant women implies a particular risk of severe deficit [10]. Several works relate 25(OH)D low levels with an increasing risk to show preeclampsia (PE), gestational diabetes mellitus (DMG), cesarean section indication, preterm birth, low birth weight (LBW) and small for gestational age (SGA).

The Vitamin D status in pregnant women is also essential for the fetus. In early pregnancy, 25(OH)D crosses the placenta from mother to fetus, and the level measured in cord blood at birth depends on maternal status being on average at 80% of the value of the mother [11]. If the mother is deficient, the same occurs to the fetus [12]. The placenta and fetal tissues express 1α-hydroxylase leading to bioactive Vitamin D in the fetal circulation.

The classic effects of Vitamin D deficiency during pregnancy and in neonates have been late hypocalcemia and nutritional rickets. Vitamin D is known to boost innate immunity by regulating production of anti-microbial peptides [13]. Several studies demonstrated that prenatal Vitamin D status plays a role in the offspring’s susceptibility to develop asthma later in life [14,15,16]. It could also contribute to the destructions of beta cells of pancreas due to its action in type 1 helper lymphocytes and cytokines [17]. Vitamin D deficit during mother’s pregnancy could also be a risk factor for multiple sclerosis in adult life because it influences early brain development, playing a relevant role in neuronal differentiation and synaptic functions [18].

In 1993, in Argentina, Oliveri et al. studied the magnitude of the problem in mothers from Ushuaia and Buenos Aires; 25(OH)D values were 6.3 ng/mL and 14.4 ng/mL, respectively [19]. Other studies, in some other countries, found a high prevalence of Vitamin D deficiency during every pregnancy trimester [20,21,22,23]. Tau et al. evaluated 25(OH)D levels in newborn infants in three cities of southern Argentina, finding 25(OH)D levels of 8.9 ± 5.7 ng/mL in babies from Río Gallegos, 12.6 ± 4.7 ng/mL in those from Comodoro Rivadavia and 14.1 ± 5.5 ng/mL in those from Ushuaia [24]. In addition to the above-mentioned study in cord blood performed in Rio Gallegos, other studies carried out in Ushuaia and Buenos Aires, at the end of winter, showed average 25(OH)D levels of 4.0 ± 2.7 and 11.3 ± 6.0 ng/mL, respectively. 25(OH)D ≤10 ng/mL levels were observed in 100% of the neonates from Ushuaia, 78% in those from Río Gallegos and 28% in those from Buenos Aires [19].

The aim of this article is to review the observational and interventional studies of the influence in Vitamin D deficiency on fertility and on pregnancy, offspring and infant outcomes. Some of the potential Vitamin D mechanisms in each outcome are also discussed.

## 2. Materials and Methods

In this review, we provide an overview of the effects of the Vitamin D and supplementation on different aspects before, during pregnancy and newborns. We also analyze the possible action of Vitamin D from preconception to infancy. Systematic reviews, meta-analyses and review articles were included in our search.

Indexed articles and articles from national journals were included. The evidence was searched in MEDLINE/PubMed, EMBASE databases and The Cochrane Library. We did not include articles that do not meet our review topic or the stated objective.

### 2.1. Vitamin D and Fertility

VDR and the enzymes involved in its activation and metabolism are highly distributed in the reproductive system of both sexes [25,26,27]. High prevalence of hypovitaminosis D is known worldwide and affects the population of any age, even young people in their fertile stage. In Argentina, low Vitamin D prevalence in young adults has a high seasonal variability, which includes a high percentage of males and females with 25(OH)D <20 ng/mL, particularly during autumn-winter seasons [28]. Several studies evaluate the role of Vitamin D and its deficiency on fertility.

### 2.2. Vitamin D and Women Fertility

At the ovarian level, Vitamin D stimulates the activity and the expression of the aromatase enzyme and induces the synthesis of estrogens and progesterone. It also increases dehydroepiandrosterone sulfotransferase transcription, while on endometrial cells, it regulates the expression of the Homeobox-A10 protein (HOXA10), which is relevant for the development of the uterus and endometrium, favoring the implantation [25,29].

Anti-Mullerian Hormone (AMH) inhibits the transition from primordial follicles to primary follicles; this regressive effect on the granulose cells is mediated by the AMH receptor (AMHR-II); thus, higher levels of AMH are responsible for suppressing the ovarian maturation. In vitro, Vitamin D decreases the mRNA AMH expression, inhibits the AMHR-II expression and increases the FSH receptor gene expression, thus indicating a positive role in follicular development and selection for ovulation [30,31]. However, those studies assessing the relationship between 25(OH)D and AMH plasmatic levels show inconsistent results [32,33,34], as in most Vitamin D actions. Plasmatic levels measurement does not probably reflect what intrinsically occurs in the tissue. A study assessing 25OHD levels in blood and follicular fluid in 53 women with infertility shows a strong negative correlation between these values (particularly with 25(OH)D >30 ng/mL levels) and the AMH levels [35].

Vitamin D action is well known in the regulation of the immune response. In endometrial cells, it reduces the inflammatory response induced by IL-1β and TNF-α and the production of toxins such as IL-6 and IFN-γ, while it increases IL-8 and TGF-β. Such actions could stimulate implantation and play a role in endometriosis. A relationship between lower levels of 25(OH)D with higher endometriosis severity and a lower risk of suffering endometriosis in women with higher levels of 25(OH)D was observed [36]. Another important result of 1.25(OH)2D3 at an endometrial level consists of inducing HOXA10 expression, a protein involved in the implantation mechanism [37].

The relationship between Vitamin D and fertility in human beings has been assessed in various studies. In couples (*n* = 132) who were searching for their first pregnancy and without any previous conditions which might upset fertility, dietary Vitamin D intake and supplements were prospectively assessed through surveys performed at baseline, 3 and 6 months. The chance to conceive was statistically higher in those couples who adhered to the daily requirement intake. Those women having sufficient levels of 25(OH)D achieved a higher pregnancy rate [38].

Various studies show that infertile women with higher 25(OH)D in plasma and/or follicular fluid levels are more likely to become pregnant after a reproduction procedure assisted by an intracytoplasmic sperm injection (ICSI) or in vitro fertilization (IVF) [39,40,41]. Other studies showed inconsistent results. As regards egg donation treatments, pregnancy rate was lower in recipient women with 25(OH)D <20 ng/mL levels versus women with 25(OH)D >30 ng/mL levels [42].

In a recently published meta-analysis, 11 studies of infertile women were included (*n* = 2455) who performed ICSI/IVF and had plasmatic measurement and/or in 25(OH)D follicular fluid. Those women having 25(OH)D >30 ng/mL obtained a higher clinical pregnancy rate [OR 1.46 (95% CI 1.05–2.02)] and a better chance of pregnancy with live newborns [OR 1.33 (95% CI 1.08–1.65)]. No differences in the number of abortions among those groups were found. The authors suggest that such results might be explained due to the Vitamin D action on the endometrium, thus encouraging the implantation [43]. In other previous meta-analyses, which included fewer studies, no differences in the clinical pregnancy rate after an assisted reproduction treatment procedure were observed in infertile women having 25(OH)D deficiency versus women with sufficient levels [44,45].

In a supplementation study with cholecalciferol (Vitamin D3) 50.000 IU weekly during 6 weeks versus placebo prior to an ICSI procedure in 85 women (32 years) with 25(OH)D <30 ng/mL, a better endometrium quality and clinical pregnancy percentage in those who were treated with supplementation (38.1% versus 20.9%) was observed, without any differences concerning oocyte response, fertilized oocytes percentage or embryo quality being found, which, again, suggests that the most important role of Vitamin D would be at an endometrial level, facilitating the implantation [46].

At the ovarian level, Vitamin D helps follicular maturation, while steroidogenesis, at the endometrium level, would encourage implantation, this probably being the main effect in fertility. As regards the assisted reproduction treatments and the ovulation induction, the results are inconsistent, and further evidence is required in order to suggest Vitamin D supplementation in patients included in this population.

## 3. Consequences of Vitamin D Deficit during Pregnancy

### 3.1. Pre-Eclampsia

PE affects 3–15% of pregnancies, it is potentially lethal for both the mother and the fetus and it is associated with intrauterine growth retardation and spontaneous or iatrogenic preterm labor. This explains its relevance in perinatal medicine [47]. Its incidence increased in recent decades, and it shows many risk factors: primiparity, PE in previous pregnancy, multiple pregnancy, PE family background, diabetes, obesity and thrombophilia [48,49].

PE is a multisystemic disease which combines high blood pressure with proteinuria. It is characterized by an abnormal maternal immune response after the implantation, which is expressed through an impaired endothelial function, activation of the coagulated cascade, increase in vascular resistance and platelet aggregation [50]. The renin angiotensin aldosterone system (RAAS) regulates blood pressure, while several epidemiological studies associated hypertension due to high renin activity with an inadequate 25(OH)D concentration because it would act like an endocrine modulator on the RAAS [51,52]. In addition to the Vitamin D endocrine suppressant effect over RAAS, it may modulate the adipokine synthesis [53], suppress proliferation of vascular smooth muscle cells [54], modulate endothelial dysfunction and delay the impaired vascular health [55].

Enzymes required for 1.25(OH)2D3 synthesis and VDR are detected in the placenta. It has been described that the 25(OH)D serum levels in women with PE differ from controls, especially in summer, unlike the expression of the genes that encode the enzymes which lack seasonal variation [56]. At the same time, an altered expression of enzymes in tissue from PE placentas was noted [57].

In 2007, Bodnar et al. described that maternal Vitamin D deficiency may be an independent risk factor for PE [58]. In 2013, a meta-analysis showed that 25(OH)D <20 ng/mL levels were associated with PE risk increase [59]. Later, Akbari et al., by analyzing 23 studies, concluded that 25(OH)D <20 ng/mL was associated with PE [60]. In another meta-analysis performed in 2022, Zhao et al. reported that a highest level of 25(OH)D was associated to a lower PE risk [OR 0.74 (95% CI 0.60–0.90)] [61].

In 2019, during the recent review of the Cochrane Library, four prospective randomized controlled trials were included, the conclusion being that Vitamin D3 supplementation reduces PE risk [RR 0.48 (95% CI 0.30–0.79)] [62]. However, the selected studies used low doses (Vitamin D3 400 IU/d and 600 IU/d in two of them) and very different dosages [63,64,65,66].

Fogacci et al. analyzed 27 randomized clinical trials including some with Vitamin D and calcium supplementation, adding 2487 women within the groups treated. Vitamin D administration was associated to lower PE risk [OR 0.37 (95% CI 0.26–0.52)] having in general a greater effect if started before the 20 weeks of pregnancy [OR 0.35 (95% CI 0.24–0.50)] and a higher supplementation dose. They concluded that the beginning of a supplementation of up to 20 weeks of pregnancy could be recommended, notwithstanding it is going to be continued up to delivery or not, being the Vitamin D3 dose around 25.000 IU/week [67].

However, the study that changed the supplementation perspective, not included in the Cochrane review, was performed by Hollis et al., who showed that to supplement Vitamin D3 4000 IU/d during pregnancy was effective, without any risks of hypercalcemia or hypercalciuria [68]. Pregnant women were administered Vitamin D3 400 IU/d, 2000 IU/d or 4000 IU/d since week 12–16 of pregnancy in order to monitor how many women of each group gave birth at the 25(OH)D target level set, which was 32 ng/mL. Previously, they had to achieve an approval from the Food and Drug Administration because of concerns about the safety of Vitamin D3 4000 IU/d. The concentrations reached were 31.6 ±14 ng/mL, 39.3 ± 14 ng/mL and 44.5 ± 16 ng/mL, respectively. The percentage of women who reached 32 ng/mL was 50% with 400 IU/d, 70.8% with 2000 IU/d and 82% with 4000 IU/d, with a decrease in the number of cesarean sections and lower PE risk [69]. The authors suggest that the current Vitamin D3 supplementation dose for pregnant women stipulated by the Institute of Medicine in 2019 had to be increased to 4000 IU/d [68,70] and that pregnancy outcomes had to be analyzed taking into account the concentration of 25(OH)D achieved instead of the dose used for supplementation [71].

### 3.2. Gestational Diabetes Mellitus

Most studies reported an inverse association between serum 25(OH)D and risk of GDM, but some studies failed to find such association.

Sadeghian et al. concluded that every 4 ng/mL increase in circulating 25(OH)D was associated with a 2% lower GDM risk, and the risk of developing GDM decreases by 29% for the highest category compared with the lowest category of 25(OH)D level [72]. Milajerdi et al., while evaluating 29 prospective and nested case–control studies, found a marginally significant positive association between 25(OH)D deficiency and GDM risk, but excluding one study with an unusual weight, pregnant women with 25(OH)D <20 ng/mL had a 26% greater risk of GDM [OR 1.26 (95% CI 1.13–1.41)] [73]. Palacios et al. concluded that supplementation with Vitamin D during pregnancy reduces the risk of GDM [RR 0.51 (95% CI 0.27–0.97)] [62]. Other studies evaluate Vitamin D supplementation in women with established GDM. In the meta-analysis by Wang et al., the supplementation significantly reduced serum fasting plasma glucose, insulin concentration and the homeostasis model assessment of insulin resistance (HOMA-IR) in women with GDM [74].

The possible mechanisms through which Vitamin D deficiency might influence the risk of GDM are not clear but include actions in beta cell and insulin resistance [75,76].

### 3.3. Cesarean Section

VDR is present in the muscle cells, either smooth or skeletal muscle cells. It has been claimed that this hormone deficit might cause a higher number of cesarean sections by reducing the strength of the contractile muscles [77].

So, 25(OH)D deficiency and insufficiency might reduce muscle mass and strength in women [78,79]. It has been described that women with 25(OH)D >30 ng/mL have a better pelvic floor performance. For women in aged 20 years or more, the risk of pelvic dysfunction was reduced about 6% for each 5 ng/mL increase in 25(OH)D and about 8.6% for women 50 years old or more [80]. During pregnancy, strengthening the pelvic floor muscles improves the muscle control, prevents urinary incontinence during and after delivery [81] and smoothens the progress of labor [82]. Vitamin D deficiency of pregnant women is likely to increase the risk of cesarean birth because of the reduction in the pelvic muscle strength and might provoke a longer and more complex labor.

Merewood et al. found an inverse association between 25(OH)D and the possibility of a cesarean section. After analyzing the logistic regression, taking into account race, age, educational level, insurance status and alcohol consumption, those women with a 25(OH)D <15 ng/mL level were almost 4 times more likely to undergo a cesarean section (28%) than those women having 25(OH)D >15 ng/mL (14%) [OR 3.84 (95% CI 1.71–8.62)] [83].

Nevertheless, the existing studies are few and contradictory. A meta-analysis about the association between 25(OH)D and cesarean section concluded that the existing reports were few and their quality of evidence was poor. Concerning supplementation, very few studies were performed, while their analysis found no clear evidence of prevention [84,85].

### 3.4. Preterm Birth and Low Birth Weight

The risk of preterm birth (PB) and low birth weight (LBW) together with SGA would increase with 25(OH)D deficiency during pregnancy [86].

Several studies state the association between 25(OH)D and PB risk. A Dutch study, by assessing 25(OH)D during the second trimester, found that those women with 25(OH)D values in the lowest quartile (<9.6 ng/mL) compared to mothers in the highest quartile (>29.5 ng/mL) had an increased risk of PB [(OR 1.72 (95% CI 1.14–2.60)], LBW [OR 1.56 (95% CI 1.02–2.39)] and SGA [OR 2.07 (95% CI 1.33–3.22)]. The estimated risk for a concentration of 25(OH)D <20 ng/mL for PB, LBW or SGA were 17.3%, 18.4% and 22.6%, respectively [87].

Another similar study showed that pregnant women with 25(OH)D <20 ng/mL at <35 gestation weeks had a 1.8-fold increased risk of PB compared with women with 25(OH)D ≥30 ng/mL, and that risk was 2.1 times higher for 25(OH)D <12 ng/mL [88].

Risk of PB was assessed in supplementation studies without finding any association with Vitamin D deficit in Asian women, so it is not clear if any differences in pigmentation and basal levels could change the results [89,90]. Cochrane’s study, by assessing seven manuscripts, did not find either prevention with supplementation, highlighting the heterogeneity and low doses used in many of them [62]. Instead, one of the studies concluded that supplementation with Vitamin D3 2.000 IU/d in women having severe deficiency reduced the risk [91]. It has also been found that VDR polymorphisms contribute to variations in Vitamin D concentrations and the increased risk of prematurity [92].

## 4. Effect of Vitamin D Status during Pregnancy in Offspring

### 4.1. Bone Parameters

Vitamin D plays a significant role in fetal skeletal growth and mineralization. Skeletal formation begins in the embryonic period, but the main period of skeletal mineralization (80%) is during the third trimester [93]. Skeletal mineralization in the uterus is primarily determined by the fetal plasma ionic calcium (Ca^2+^) concentration, which is dependent on placental Ca^2+^ transfer and fetal calciotropic hormones. The level of Ca^2+^ transport across the placenta is strongly regulated by plasma membrane calcium-dependent ATPases (PMCA 1–4) gene expression [94].

The expression of the PMCA3 mRNA predicts neonatal whole body bone mineral content (WB-BMC) at birth [95], and there is some experimental evidence that PMCA gene expression might be influenced by 1.25(OH)2D [96]. The effect of Vitamin D on skeletal development might be mediated, at least in part, through placental calcium transport and the bioavailability of calcium to fetus [22]. In the last decades, an interesting number of studies has detected a relationship between Vitamin D nutritional status in pregnancy or in cord blood and offspring’s parameters of bone mass, quality and size studied by different techniques: DXA, ultrasound (US) and peripheral quantitative computed tomography (pQCT) [97,98,99,100].

A three-dimensional US of femoral morphology performed at 34 weeks of pregnancy showed a positive relationship between 25(OH)D levels and femoral volume [99]. Lower WB-BMC, adjusted for weight by DXA, was observed in neonates with 25(OH)D <13.2 ng/mL and in 15 day infants born to mothers with 25OHD <15 ng/mL compared to those with higher levels [98,101]. Studies with pQCT of tibia found higher tibial bone mineral density (BMD), bone mineral content (BMC) and cross-sectional area in neonates with 25(OH)D 20.8 ng/mL levels than in those with <14.5 ng/mL levels [100].

Other studies found no relationship between bone parameters assessed by DXA: WB-BMD, BMC, bone area (BA) in the first month of life and 25(OH)D measured during pregnancy [102,103,104,105].

Most of the studies that showed significant differences in bone parameters after birth were those with 25(OH)D levels in the range of deficiency (<20 ng/mL) and even severe deficiency (<10 ng/mL), which would lead to the conclusion that mineralization would be fundamentally affected in fetuses with very decreased levels of 25(OH)D.

Another topic of interest was whether this difference in intrauterine mineralization can be maintained throughout childhood and adolescence and favour a higher peak of bone mass that could be associated with lower risk of fractures in the rest of life [106,107].

Javaid et al. found significantly lower WB-BMC in the offspring (at 9 years old) of women who were Vitamin D deficient (25(OH)D <11 ng/mL) compared to those of women with 25(OH)D > 20 ng/mL during late pregnancy in 198 mother-and-child pairs [22]. Zhu et al. reported lower WB-BMC and WB-BMD in the offspring of women with Vitamin D deficiency (25(OH)D <20 ng/mL) during mid-pregnancy in 341 mother-and-child pairs [108].

By contrast, studies involving high numbers of participants found no relationship with 25(OH)D levels in pregnancy and measurements of bone mass in childhood. A large cohort of 3960 mothers–children (average age 9.9 years) of the ALPSAC protocol found that there was no difference in WB and LS. BMC, areal BMC or BA in children whose mothers had Vitamin D deficiency (<11 ng/mL) or insufficiency (11–20 ng/mL) compared to those with sufficiency (>20 ng/mL) during first, second or third trimester of pregnancy [109]. The Generation R Study (*n* = 3034) also found no association between 25(OH)D levels, measured at week 20 of pregnancy, and offspring BMD in 6-year-old children [110].

The difference in the results of these observational studies emphasized the need for high-quality randomized controlled protocols to clarify the effect of Vitamin D status and/or supplementation in bone parameters at birth and in childhood.

There are some recently published studies of Vitamin D supplementation in pregnancy that evaluate offspring bone mass, of which the largest is the Maternal Vitamin D Osteoporosis Study (MAVIDOS). This is a randomized double-blind placebo-controlled trial of pregnancy Vitamin D3 supplementation from 14 weeks of gestation until delivery. The primarily goals were to assess offspring bone mass at birth and early childhood. A total of 1134 women were randomized to Vitamin D3 1000 IU/d or placebo; 965 remained in the study until delivery. Blood samples were taken at 14 and 34 weeks of gestation and from cord blood. Maternal and offspring (*n* = 736) DXA scans were performed within 2 weeks of birth. Follow up of the offspring is planned at regular intervals up to 6 years, including DXA, pQCT, High-resolution pQCT and hand grip strength [111,112,113].

In total, 83% of women randomized to Vitamin D3 achieved a 25(OH)D >20 ng/mL in late pregnancy, compared with only a 36% in the placebo group. Although there were no differences in WB-BMC, BA or areal BMD between the two groups in general, the children born in winter–early spring to mothers randomized to supplementation showed WB-BMC and BMD approximately 9% and 5% higher, respectively, with a 0.5 standard deviation and a higher increment in neonatal WB-BMC compared to those born from mothers in the placebo group. This effect size is substantially larger than those observed between children with and without fractures and, hence, if persisting into later childhood, is likely to be clinically relevant [114].

The Copenhagen Prospective Studies on Asthma in Childhood (COPSAC2010) compared Vitamin D3 daily supplementation of 2800 IU/d versus 400 IU/d as control group, which was started in mid-pregnancy until one week after birth. In total, 517 6-year-old children underwent DXA scans. Children that received Vitamin D3 2800 IU/day had higher WB (less head), BMC and areal BMD. The largest effect was observed in children from Vitamin D insufficient mothers (25(OH)D <20 ng/mL) and among winter births [115].

In contrast to the positive findings of MAVIDOS and COPSAC trial, the findings of two small studies (*n* = 52 and *n* = 25) which assessed Vitamin D3 daily doses of 2000 IU versus 400 IU or Vitamin D3 weekly doses of 60.000 IU (4 or 8 weeks) versus placebo should be mentioned. None of the assessments of DXA scans performed in offspring at first month [116] or in infants at 12–16 months [117] showed any difference in infant bone mineralization in response to antenatal Vitamin D supplementation.

In conclusion, the supplementation with Vitamin D3 2800 or 1000 IU/d during pregnancy showed positive findings in offspring bone mass. However, it is clear that 1000 IU/d do not put an end to the seasonal variation in 25(OH)D during late pregnancy, considering that approximately 22% of the supplemented women (in winter–early spring) still have a 25(OH)D level <20 ng/mL. Regarding safety, no adverse effects were reported in these trials and even at higher doses up to Vitamin D3 4000 IU/d [68].

A possible mechanism of the effect of Vitamin D supplementation in pregnancy and offspring bone mass is the fetal programming that refers to the mechanism whereby environmental effects during critical periods of early development lead to persistent changes in structure and function, which may be mediated by epigenetic mechanisms of imprinted genes that regulate fetal and placental growth [118]. Epigenetic changes could be the link between the nutritional Vitamin D status of pregnant women and the beneficial effects in offspring bone mass. Changes in methylation of two loci in umbilical cord DNA have been described: CDKN2A (with a possible role in skeletal growth and bone cell activity) and the retinoid receptor alpha (RXRA) gene that forms a heterodimer with the VDR were inversely related to weight-adjusted BMC in 4-year-old infants [119,120,121].

On the other hand, Vitamin D can induce epigenetic changes in several genes involved in Vitamin D metabolism, although this was not confirmed in other publications [122,123,124].

### 4.2. Bronchiolitis

Lower respiratory tract infections (LRTIs), primarily pneumonia and bronchiolitis, are a leading cause of morbidity and mortality in early childhood. Vitamin D is known to boost innate immunity by regulating production of anti-microbial peptides [13]. Furthermore, the active form of Vitamin D, calcitriol, is produced locally in the lung epithelium, and its immunomodulatory properties have been shown to play a role in host defense against respiratory tract pathogens [125].

A prospective study of 156 neonates in the Netherlands reported an increase in respiratory syncytial virus associated LRTI in deficient newborns with 25(OH)D <20 ng/mL compared with 25(OH)D >30 ng/mL [126]. Another study on Saudi infants demonstrated that low 25(OH)D was associated with increased risk of LRTI in the first 2 years of life, with lower concentrations occurring in infants who developed respiratory infections compared with those who did not: 13.5 ± 1.16 ng/mL versus 28.5 ± 1.08 ng/mL (*p* < 0.0001) [127].

In 2014, Łuczyńska et al. observed a statistically significant association between 25OHD status in cord blood and risk of LRTIs in a cohort of 777 mother–infant pairs [128]. The adjusted RR of LRTI for individuals with Vitamin D deficiency (<10 ng/mL) in comparison to the referent category (>20 ng/mL) was 1.32 (95% CI 1.00–1.73). The effect was stronger in infants born in autumn [RR 3.07 (95% CI 1.37–6.87)].

Other studies found lower levels of 25(OH)D in infants with bronchiolitis and a significant inverse correlation between serum 25(OH)D levels and disease severity [129,130]. Alakas et al. studied 182 children with acute bronchiolitis in Turkey. Vitamin D deficiency was closely linked to severe bronchiolitis and the need for intensive care unit admission in infants [131]. Vo et al., in a large, multicenter cohort of 1016 US infants (mean age 3.2 months) hospitalized with bronchiolitis, observed that infants with 25(OH)D <20 ng/mL had increased risk of intensive care [(OR 1.72 (95% CI 1.12–2.64)] and longer length-of-stay [adjusted RR 1.39 (95% CI 1.17–1.65)] compared with infants with 25(OH)D ≥30 ng/mL. [132].

A double-blind, randomized clinical trial was performed on 89 infants with bronchiolitis. Patients were randomized to receive Vitamin D3 (100 IU/kg, range 500–1300 IU) daily supplementation or placebo [133]. The intervention group had a significant improvement in three parameters over the placebo group: the mean time taken for resolution of the disease, the mean time taken for the improvement of oral feeding and the duration of hospitalization.

### 4.3. Asthma

Several studies have suggested that Vitamin D supplementation during pregnancy could reduce childhood asthma rates [16]. Camargo et al. found that a higher maternal intake of Vitamin D during pregnancy may decrease the risk of recurrent wheeze in children at 3 years old [134]. In the VDAART study, pregnant women were supplemented with Vitamin D3 4400 IU/d from 10–18 weeks of gestation until delivery, with an evaluation of prevention of asthma/wheeze in the infant/child at 1, 2, 3, and 6 years post-birth. The incidence of asthma and recurrent wheezing in their 3-year-old children was lower by 6.1%, although it was not significant. A post-hoc analysis of VDAART described that in women entering pregnancy with circulating 25(OH)D >30 ng/mL who were prescribed Vitamin D3 4000 IU/d starting at approximately 10–18 weeks of gestation were associated with the lowest risk for asthma/recurrent wheeze at 3 years of age compared with those having 25(OH)D <20 ng/mL and receiving placebo [15,135].

These data suggest that Vitamin D is strongly associated with very early in utero lung development in the fetus that cannot be reduced by starting Vitamin D supplementation at the end of the first trimester. Vitamin D related genes in early lung development are associated with asthma pathogenesis [14].

Mothers with asthma had 2 times higher risk of having a child with asthma or recurrent wheeze before the age of 3 years than did mothers without asthma. However, this risk among mothers with asthma was substantially attenuated if they were Vitamin D sufficient at early and late pregnancy. Therefore, women with asthma who start their pregnancies with high levels of Vitamin D and remain Vitamin D sufficient throughout pregnancy are likely to experience a reduced risk of asthma or recurrent wheeze in their children before 3 years of age [136].

Grant et al. demonstrated that Vitamin D3 supplementation during pregnancy and childhood reduces aeroallergen sensitization. They supplemented woman–infant pairs from 27-week gestation to birth with placebo or 1 or 2 dosages of daily oral Vitamin D3, placebo/placebo, 1000 IU/400 IU or 2000 IU/800 IU, and specific IgE was measured in children at 18 months. Vitamin D3 reduced the proportion of children sensitized to mites at the age of 18 months [137].

### 4.4. Type 1 Diabetes

High doses of Vitamin D early in life may help prevent type 1 diabetes (T1D) [138]. Hyppönen et al. supplemented 2000 children in Finland with Vitamin D3 2000 IU/d during the first year of life, with a decrease in the incidence of T1D during a 30-year follow-up [139].

Makinen et al. showed that the increased 25(OH)D concentrations observed since 2003 in Finnish children had a delayed temporal association with the reversal of the rising trend in the incidence T1D after 2006 [140]. Sørensen et al., using data of Norwegian Childhood Diabetes Registry, reported the association of lower levels of 25(OH)D during the third trimester in pregnant women delivering a child who developed T1D before the age of 15 years [141]. A similar study did not achieve the same result [142]. In 2018, Thorsen et al. reported no association between maternal 25(OH)D and risk of T1D, with blood drawn in weeks 7–9 and 24–25 of pregnancy in Danish women and in weeks 17–18 in Norwegian women [143].

Higher maternal DBP level at delivery may decrease offspring T1D risk. DBP has been reported to be important in production of the antimicrobial peptide cathelicidin in monocytes by regulating the bioavailability of 25(OH)D [144]. Low DBP levels toward the end of pregnancy could influence antimicrobial response and inflammation in the mother, which could predispose for offspring autoimmunity.

The inconsistent results concerning 25(OH)D levels and T1D risk mentioned above [8,33,34] could be explained in part by genetic causes, and a relationship with polymorphisms of VDR has been described, particularly with rs11568820. A large study, instead, did not find such association [6], although these data may depend upon the 25(OH)D level [42]. These findings would support the premise that maintaining an adequate level of Vitamin D in fetal life could have the potential to prevent T1D.

### 4.5. Multiple Sclerosis

Risk of Multiple Sclerosis (MS) is multifactorial, as genetic and environmental factors have been hypothesized. MS is more frequent with increasing latitudes, and patients are more likely to be born in spring months [18,145,146].

In 2016, Munger et al. studied whether serum 25(OH)D levels in early pregnancy were associated with risk of MS in offspring. Deficient maternal 25(OH)D levels during pregnancy (<12 ng/mL) were associated with a 90% increased risk of developing MS as an adult [RR 1.90 (95% CI 1.20–3.01)], with a 43% reduced risk of MS associated with every 4 ng/mL increase in maternal 25(OH)D level [RR 0.57 (95% CI 0.28–1.18)]. The average age at MS diagnosis was 19.8 and the oldest age at diagnosis was 27 years. Deficient pregnant women had a 43% higher MS risk [RR 1.43 (95% CI 1.02–1.99)] as compared to women with 25(OH)D levels ≥20 ng/mL [147].

### 4.6. Autism

Autism spectrum disorders are a group of disorders characterized by a persistent disturbance in interpersonal interaction, with restrictive or repetitive patterns of behavior and verbal and non-verbal communication. The pathogenesis is not clear. The evidence shows that genetic factors play a role in its occurrence, but non-genetic factors are probably involved in its pathogenesis. The prevalence has been estimated at 1.5%, and it has been reported that it is increasing, which poses the question of whether this increase is real or due to a greater diagnosis [148].

In 1995 and 1996, a Swedish study noted that the incidence of autism was much higher among children from the Ugandan immigrant community compared to Swedish [149], and in 2008, it was established that Swedish children had a prevalence of 0.19% while it was 0.70% in the children of the Somali community from the same country. It was considered whether a decreased level of 25(OH)D could be a risk factor for this pathology.

In 2008, a review outlined the hypothesis that Vitamin D deficiency played an important causal role in this pathology [150,151]. It was shown that more autistic children are born in winter and spring in northern Europe and the United States but not in regions such as California and Israel, where there is good solar irradiation throughout the year.

Numerous association studies have been published that confirmed that autistic individuals have a lower level of 25(OH)D than their siblings. A meta-analysis with 11 studies showed lower 25(OH)D in patients than in controls. In addition, this has been already discovered in samples taken at birth [152]. These data rule out that “lifestyle” is the cause of the deficiency found.

Several mechanisms could explain this relationship. Firstly, Vitamin D influences the early brain development in children. It plays a role in neuronal differentiation, neurotransmission and synaptic functions. Secondly, Vitamin D deficiency may alter T cell activation profile and affect the adaptive immunity and cause preponderance to autism [153]. In addition, oxidative stress may increase the susceptibility to autism by their lethal interaction with genetically susceptible genes. Serotonin plays an essential role in controlling emotions. Vitamin D increases the synthesis of cerebral serotonin (but not peripheral) by stimulating the enzyme tryptophan hydroxylase type 2 [154]. Lastly, Vitamin D deficiency could increase the risk of genetic mutations by inhibiting DNA repair of early mutations and thus could contribute to the occurrence of autism [148].

There are few studies with Vitamin D supplementation in autistic children, pointing out that of Feng et al. with 37 children who completed 3 months of Vitamin D supplementation and evaluating them with The Autism Behaviour Checklist and the Childhood Autism Rating Scale scores, finding improvement especially in the subgroup of children under 3 years of age [155].

An interventional study recently reviewed the benefits of Vitamin D supplementation in pregnancy to decrease the incidence of autism in the offspring. Pregnant women with a previous autistic child were supplemented with Vitamin D3 5000 IU/d, followed by the supplementation of a newborn with Vitamin D3 1000 IU/d for the first 3 years of life. These children were followed up at 18 and 36 months of age. The results were promising, with only 1 out of 19 children developing autism (5%), compared to the general recurrence risk of 20%. This study suggests that Vitamin D3 supplementation in pregnancy could reduce the risk of autism in children, but more studies are warranted to confirm the conclusion [156].

## 5. Conclusions

Worldwide, the prevalence of hypovitaminosis D is high before, during and after pregnancy. Winter months, together with a high body mass index, are considered weight risk factors in order to find low levels of 25(OH)D during pregnancy. Since before becoming pregnant Vitamin D would seem to be important due to its potential role in fertility. To date, during pregnancy, different studies have shown the association between hypovitaminosis D and the onset of preeclampsia, cesarean section indication, preterm delivery, low birth weight, low weight for gestational age and gestational diabetes. The effect of the intervention shows positive changes in the bone mass of the newborn and promising results in the prevention of respiratory infections such as bronchiolitis, the development of asthma, delay in the onset of type 1 diabetes, multiple sclerosis and autism and pathologies in the newborn, which are associated with low Vitamin D during pregnancy. As a study group, we recommend reaching an optimal level of 30 ng/mL or more before conception and throughout the entire pregnancy with different dosages available, avoiding levels below 20 ng/mL.

From the analyzed review, we found that for the non-classical actions of Vitamin D, values greater than 40 ng/mL were shown to prevent infectious diseases, predominantly respiratory, and effects on autoimmune diseases, with which the intervention would favor prevention.

Evidence allows proposing the substitution since before conception, during pregnancy and in the first months of life, although more observational and interventional studies, including a large cohort of subjects during preconception, pregnancy and in infants, are required.

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
