# Peer review of "Vitamin D: Before, during and after Pregnancy: Effect on Neonates and Children"

_nutrients, 2022, doi:10.3390/nu14091900_

Round 1

Reviewer 1 Report

Thank you for asking me to review this paper. In this paper the authors discuss the role of vitamin D before, during and after pregnancy.

This paper is well written and covers most aspects of pregnancy. It also covers the role of vitamin D in children (autism, asthma etc). I would therefore suggest a change in title to cover this aspect of the review.

Although it covers some aspects of vitamin D in pregnancy it would be more complete to look at the role of supplementation in each aspect of pregnancy and newborns and children.

The strength of each of the studies mentioned (grading) would be helpful.

There are a lot of grammatical and spelling errors. A thorough check for these is recommended.

Please mention 25OHD as 25(OH)D

VD3 is not standard term used.

At the start of the paper the authors would need to mention their search criteria (as well as inclusions and exclusions if any).

Author Response

Thank you, we changed the title, the terms 25OHD and VD3. We mention the search criteria and change abstract and conclusions.

Reviewer 2 Report

The manuscript focuses on the effect of vitamin D on fertility, antenatal adverse outcomes and fetal bone health and related diseases after birth. Meanwhile, the potential mechanism of vitamin D on relevant outcomes is mentioned also. However, authors mainly described the findings from the related studies, but did not deeply conclude the implications of related studies. Finally, this manuscript can’t provides the guiding effect to readers. Additionally, The abstract must cover the main contents and conclusion of this topic. There are some typo errors in the manuscript, such as: Line 237, “250HD” and Line 240, “5 Ing/mL” should be “25OHD” and “5 Ing/mL”, respectively.

Author Response

We change abstract and conclusion and we correct the errors in manuscript.

Round 2

Reviewer 1 Report

I’ve read the paper and although they have made improvements there are still a lot of grammatical errors and spelling ERRORS. Also in some places they say vitamin D and in others VD. I do not like term VD.  Also when they use abbreviations first time they should give the full form first. 

Reviewer 2 Report

This version is good, but there are some typo errors in the current version, such as:

  1. P1, in the sentence “which is hydroxylated in the liver at 25-hydroxyvitamin D 25(OH)D”, “25-hydroxyvitamin D 25(OH)D” would be add a parentheses “25-hydroxyvitamin D (25(OH)D)”.
  2. P2, In the sentence “It could also contribute to the destructions of beta cells of pancreas due tb go its action in type 1 helper lymphocytes and cytokines (17)”, “due tb” would be “due to”.